# Animal Protein Intake Is Associated with General Adiposity in Adolescents: The Teen Food and Development Study

**DOI:** 10.3390/nu12010110

**Published:** 2019-12-31

**Authors:** Gina Segovia-Siapco, Golandam Khayef, Peter Pribis, Keiji Oda, Ella Haddad, Joan Sabaté

**Affiliations:** 1School of Public Health, Loma Linda University, Loma Linda, CA 92350, USA; koda@llu.edu (K.O.); ehaddad@llu.edu (E.H.); jsabate@llu.edu (J.S.); 2Don B. Huntley College of Agriculture, California State Polytechnic University, 3801 West Temple Avenue, Pomona, CA 91768, USA; 3Department of Individual, Family & Community Education, Nutrition and Dietetics Program, College of Education, University of New Mexico, Albuquerque, NM 87131, USA; pribis@unm.edu

**Keywords:** adolescents, protein, obesity, central adiposity, waist-to-height ratio, fat mass, animal protein, dietary assessment, web-based food frequency questionnaire, vegetarian

## Abstract

Protein plays a crucial role in the growth and development of adolescents. However, being a secondary energy source, protein’s role in obesity has been sidelined. We examined whether intake of protein (total, animal, plant), branched-chain (BCAAs), and sulfur-containing (SCAAs) amino acids are associated with general body and central obesity and body composition in a cross-sectional study among healthy adolescents. Students aged 12–18 years old (*n* = 601) in schools near two major Adventist universities in California and Michigan provided dietary data via a validated web-based food frequency questionnaire (FFQ) and anthropometric data during school visits. Intakes of total, animal, and plant proteins, and BCAAs and SCAAs were derived from FFQ data. We defined general body obesity with body-mass-index-for-age (BMIz) z-scores and central obesity with waist-to-height ratios (WHtR). After full adjustment for covariates, multiple regression analyses showed significant positive associations between intakes of total protein (β = 0.101, 95% CI: 0.041, 0.161), animal protein (β = 0.118, 95% CI: 0.057, 0.178), BCAAs (β = 0.056, 95% CI: 0.025, 0.087), and SCAAs (β = 0.025, 95% CI: 0.012, 0.038) with general body adiposity. Animal protein (β = 0.017, 95% CI: 0.001, 0.033) and SCAAs (β = 0.004, 95% CI: 0.000, 0.008) were also associated with central obesity. Total and animal protein and BCAA and SCAA were also significantly associated with fat mass. Our findings suggest that high protein intake may pose a possible detriment to adolescent health. Longitudinal and safety evaluation studies are recommended.

## 1. Introduction

The increasing rates of obesity are a global problem and continue to be a health hazard, in particular, among the youth. In the United States, about 21% of adolescents are obese according to the 2015–2016 NHANES report [1], while about 33% of 6- to 18-year-olds are overweight [2]. Moreover, about 33% of 6- to 18-year-olds in the United States have been classified as abdominally obese [3]. These statistics are particularly concerning given that body composition in adolescence often tracks into adulthood, thereby increasing the odds of remaining overweight or obese [4] or increasing cardiometabolic disease risk [5,6]. Obesity, particularly abdominal/central obesity, which reflects visceral fat accumulation, is associated with mortality risk [7] and several comorbidities [8,9,10,11,12]. The development of obesity is multifactorial [13]: In general, imbalances in the diet, e.g., energy-dense diet. Several investigations on factors that contribute to childhood and adolescent obesity have focused on dietary factors [14,15,16,17,18] or habits [19,20,21,22,23]. One macronutrient that is particularly critical for normal growth and development of adolescents is protein [24], but its role in obesity development is still controversial. Although a chief actor within every cell in the human body [24,25], protein intake recommendations at different life stages—in both amounts and sources/types of protein—have not been firmly established over time, perhaps due to differing theories amongst scientific community members themselves. While most plant foods contain protein, the majority of U.S. adults consume their recommended amounts mostly from animal sources (i.e., meat, fish, poultry, and dairy), while grains serve as the major plant source [26]. Among adolescents, protein foods that contribute mostly to energy intake are also of animal origin [27]. Findings on the role of protein in obesity are highly varied in studies among adults [28,29,30,31,32] and children/adolescents [33,34,35,36]; however, data are still limited among adolescents.

Similar to reports on animal protein and adiposity, increases in plasma concentration of the branched-chain amino acids (BCAAs: Leucine, isoleucine, and valine) in response to overnutrition—particularly in the context of high fat diets—have been linked to pediatric obesity and obesity-related insulin insensitivity, increasing the risk of type 2 diabetes later in life [37]. For instance, elevations in plasma concentration of BCAAs were significantly associated with higher body-mass-index-for-age (BMIz) z-scores in a cross-sectional cohort of 8- to 18-year-old healthy subjects [38]. However, some argue that plasma concentration of BCAAs is not a direct reflection of diet-derived BCAAs [39]. In fact, one study reported inverse associations between dietary intake of BCAAs and overweight/obesity in healthy middle-aged adults from East Asian and western countries [40].

Less controversial in their link to adiposity and yet relatively novel in the world of protein research are the sulfur-containing amino acids (SCAAs), methionine and cysteine. In animal studies, increasing cysteine intake substantially increases body weight gain [41], while restriction of dietary methionine, the precursor of cysteine, decreases visceral fat mass [42,43]. Large observational human studies have reported similar results to animal studies, though the evidence for association with adiposity is more robust for plasma cysteine concentration rather than dietary cysteine [43,44]. No current report exists on dietary intake of these amino acids in healthy adolescent populations and their potential links to body weight and composition. Given that many dietary sources of animal protein are also high in dietary BCAAs and SCAAs, we put these associations to the test in a sample of healthy adolescents who follow a relatively healthy lifestyle and consume a variety of plant and animal protein foods.

Perhaps protein’s role in obesity development had been put on the sideline due to the fact that this macronutrient is essential for growth and development. Although there is still a dearth of studies among adolescents, evidence is growing that intakes of protein—particularly the type of protein eaten—and certain amino acid groups are linked to overweight/obesity. Thus, we examined if overweight/obesity in a healthy adolescent population known to espouse smoke-free and alcohol-free living was associated with their intake of protein (plant and animal proteins) and two groups of amino acids (the BCAAs and SCAAs). Secondarily, we also determined if the same nutrients were also associated with body composition, particularly fat mass and fat-free mass.

## 2. Materials and Methods

### 2.1. Study Design and Participants

The Teen Food and Development Study (TeenFADS) is a cross-sectional study that was designed to explore potential associations between intake of different types of food and physical growth and pubertal development in adolescents. Details of the study and sampling method have been described elsewhere [45]. Briefly, participants were recruited from schools near major Adventist universities in Southern California and Michigan. A total of 601 adolescents (339 females and 262 males), aged 12–18 years old (grades 7–12), completed a self-administered web-based survey that included a dietary assessment section and sections on physical development, lifestyle habits, and demographics. Anthropometrics (weight, height, waist and hip circumferences, and body composition) were measured during school visits. A similar number of adolescents from California (*n* = 301) and Michigan (*n* = 300) participated. The Institutional Review Boards of Loma Linda University (IRB #5120014) and Andrews University (IRB Protocol #12-113) approved all the study protocols. Parents of adolescents younger than 18 years old provided permission for their children to participate, while participants themselves provided assent.

### 2.2. Assessment of Dietary Intake

The dietary assessment section of the web-based survey was a validated 151-item semiquantitative food frequency questionnaire (FFQ) [46]. The FFQ consisted of food items categorized into convenience foods (32 items), protein-rich foods (29 items), starches/cereals (17 items), vegetables/fruits (21 items), dairy products (10 items), beverages (24 items), snack sweets (11 items), and soups/legumes (7 items). Frequency of intake categories were: Never/rarely, 1–3 times per month, once per week, 2–4 times per week, 5–6 times per week, once per day, 2–3 times per day, ≥4 times per day, and seasonal fruits, which were counted as 1–3 times per month.

#### Assessment of Protein and Amino Acid Groups Consumption

For the purpose of this study, protein foods were categorized according to the following protein sources: Red meat, white meat/poultry, processed meat, fish, dairy, egg, grain, gluten, soy, non-soy legumes, and nuts/seeds/nut butters. Appendix A presents the protein food categories and the corresponding FFQ food items under each category. Intake frequencies were converted to frequency of intake per day, which is essentially equivalent to the number of servings per day as the FFQ is semiquantitative. The Nutrition Data Systems for Research (NDS-R) version 2012 database developed by the Nutrition Coordinating Center (NCC) [47] was used to determine grams of protein per serving of food and the protein and amino acid composition of the foods. Total protein and amino acid intakes were calculated using the product sum method [48]. Protein sources and the contribution of each protein type to the total animal or plant protein were determined from all the foods except fruits and vegetables. Foods containing a single ingredient were readily assigned a protein type and source whereas the process involved more steps for mixed foods (e.g., burritos and meat sandwiches). Mixed foods were broken down into their component foods, and the protein contribution was determined for each food component and categorized according to protein type. For foods with no recipe information, in particular meat alternatives, the proportions of ingredients and therefore protein types, e.g., soy and gluten, were estimated according to the order in which they appeared on the food label ingredient list (e.g., soy isolate, soy protein, or gluten extract for veggie burger).

Total protein intake was measured in grams per day (g/day), in grams per kg body weight per day (g/kgBW/day), and as % total caloric intake, while animal protein and plant protein intakes and their food sources were measured in g/day, as % total protein, and as % total caloric intake [49]. Intake of BCAAs was determined by taking the sum of the total intakes of the branched-chain amino acids leucine, isoleucine, and valine. SCAA intake was measured as the sum of the total intake of the sulfur-containing amino acids, methionine and cysteine.

### 2.3. Assessment of Indicators of Overweight, Obesity, and Body Composition

Trained staff took anthropometric measurements during school visits/clinics. They measured height, weight, and waist and hip circumferences two times for precision, and the average of the two measures was used. Weight and body composition (fat mass and fat-free mass) was measured with a bioelectric impedance analysis scale, TANITA™, and height was measured using a stadiometer. Waist and hip circumferences were measured with a nonstretchable cloth measuring tape.

BMIz scores were used as the measure of general body obesity, while waist-to-height ratio (WHtR) was used to measure central adiposity. For body composition, fat mass and fat-free mass were used. BMIz scores were calculated using the World Health Organization’s growth references for ages 5 to 19 years [50]. Participants were categorized according to the following definitions: Normal = BMIz < 0.99 and overweight/obese = BMIz ≥ 1.0. WHtR is considered a more accurate measure of abdominal or central obesity among adolescents [50]; thus, it was used instead of waist circumference. The cutoff point of 0.5 was used to differentiate between normal and abdominally obese [51,52,53].

### 2.4. Assessment of Other Variables

The original questionnaire also included demographic questions including age, gender, child’s ethnicity (determined based on parents’ ethnicities), parental educational levels, and study site (California vs. Michigan). Some lifestyle questions included physical activity per day and the number of hours of sleep per night. Vegetarianism was determined using an a priori definition of consuming no more than one combined serving of meat/meat derivatives, fish, and poultry per week.

### 2.5. Data Analysis

Seventy subjects were excluded from analyses due to either improbable caloric intake (defined as <500 kcal/day or >3500 kcal/day for girls and <800 kcal/day and >4500 kcal/day for boys). This left a total of 530 (299 girls and 231 boys) in the final analytical dataset. Due to missing anthropometric measures, another 11 were excluded in several analyses that used BMIz and WHtR, and in those cases, the sample size was down to 519 participants. Descriptive analyses were performed using the statistical software SPSS for Windows version 22 (IBM SPSS, Inc.). Given the near-normal distribution of the dietary exposure variables (protein and amino intakes), parametric comparison tests were used.

To investigate if any relationships existed between protein/amino acid intake and obesity, we used multiple regression. Outcomes were BMIz, WHtR, fat mass, and fat-free mass. We independently looked at height-for-age z-scores (Htz) to find out whether linear growth was associated with protein intake in this group of adolescents. Distributions of the outcome variables were examined for skewness and outliers by histograms and normal probability plots. If any deviation from normality was suspected, we applied either log or inverse transformation to normalize the data. Fat mass and fat-free mass were log-transformed and WHtR was inverse-transformed. No transformation was necessary for BMIz and Htz.

For each of the outcomes, we ran regression models that included each of the following as the predictor/independent variables: (1) Total protein, (2) animal protein, (3) plant protein, (4) BCAAs, and (5) SCAAs. Separate analyses were also done for cysteine and methionine. All proteins and amino acid groups were adjusted for total energy intake by the residual method prior to regression analysis. To control for confounding, age (in years), gender, ethnicity, site, and total energy served as covariates in the “base” model. The “full” model further adjusted for total fat or total carbohydrates, hours of physical activity, and hours of sleep per day in addition to the base model variables. For animal protein, plant protein was included in the model, while for plant protein, animal protein was included. Scatter plots and variance inflation factors were examined for multicollinearity among the covariates/independent variables. Adherence to regression assumptions was verified by visual inspection of residual plots. Results were presented as beta coefficients (β) for exposure variables, 95% confidence intervals, and *p*-values. All regression analyses were conducted in R version 3.2.4 [54].

## 3. Results

### 3.1. Demographic Characteristics of Participants

Table 1 presents the characteristics of the TeenFADS participants based on the 530 participants who were included in the analyses. About 56% were females, overall mean age was 15.0 (SD = 1.7) years, and approximately 76% of adolescents were in the older age category (i.e., 14–18 years). Over one-third of the participants were Caucasian, and the majority had parents with some level of college education and higher, with less than one-fifth of parents having educational levels of high school or less. Twenty-six percent of all participants were vegetarians (defined in this study as individuals consuming less than one combined portion of meat, meat derivatives, poultry, and fish per week) with girls being more likely to be vegetarians. Means for BMIz were within the normal range, yet about 22% of adolescents were categorized as overweight/obese (using cutoff points of BMIz < 0.99 for normal or BMIz ≥ 1.0 for overweight and obese). Slightly more girls were overweight/obese with a borderline significant BMIz of 0.34 (SD = 0.88), compared to boys with a BMIz of 0.17 (SD = 1.1). About 20% had abdominal obesity (using a WHtR cutoff point of <0.5 for normal and ≥0.5 for abdominally obese) with significantly more girls having abdominal obesity. As expected, girls had a higher mean for fat mass and a lower mean for fat-free mass compared to boys. In addition, but not shown in the table, the older age group (14–18 years) had a higher mean BMIz compared to the younger age group (12–13 years) for both genders, whereas WHtR was consistent across age groups.

Table 2 presents relevant characteristics of participants according to categories of BMIz and WHtR. Adolescents categorized as overweight/obese in both categories of general body weight and central obesity had a higher intake of fat and lower intake of carbohydrates, shorter time spent on vigorous physical activity, and higher fat mass and fat free mass compared to their normal-weight counterparts.

### 3.2. Dietary Caloric, Protein, and Amino Acids Intake

For all participants, mean total protein intake accounted for 16% of total energy intake with nearly 45% coming from animal protein and 55% from plant protein (see Table 3). Males had a significantly higher intake of total protein, both in absolute values (g/day) and as a proportion of total energy intake, and higher absolute intakes of BCAAs and SCAAs. Table 3 also shows that the proportion of energy intake from protein was significantly higher among the overweight/obese group (16.5% compared to 15.8% in the normal weight group). Intake in g/day and as % total protein was lower for animal protein, but higher for plant protein among the normal weight group compared to their overweight/obese counterparts. Absolute intakes of BCAA and SCAA, as well as the SCAA proportion of total protein intake, were also lower in the normal weight group. Those with normal WHtR had a relatively higher intake of plant proteins (~47 g/day or 56% of total protein intake) compared to those with obese WHtR (~43 g/day or ~52% of total protein intake). The proportion of SCAA in total protein intake was also lower in the normal WHtR group, 3.4% vs. 3.5%, (see Table 3).

### 3.3. Protein Food Sources and Their Contribution to Total Protein

Table 4 shows that, overall, dairy products contributed the most (~18 g/day, ~25%) to total protein intake, followed by grains (~17 g/day, ~22%). Soy protein contributed about 15% (~10 g/day) to total protein intake, and red meat contributed about 10% (~7 g/day), followed by poultry (~6 g/day, 8.5%), non-soy legumes (3.6 g/day, ~5%), nuts (3.4 g/day, 4.7%), egg (2.6 g/day, 3.6%), gluten (2 g/day, 3%), fish (1.2 g/day, 2%), and processed meat (1 g/day, 1.5%). Adolescent boys had higher intakes of red meat and grains than girls. Overweight/obese adolescents consumed substantially higher amounts of both red meat and poultry and lower amounts of soy and non-soy legumes compared to the normal weight group. Intakes of soy and non-soy legumes were significantly lower in the abdominally obese group.

### 3.4. Associations between Intake of Protein and Amino Acids and Obesity and Body Composition

Table 5 shows that higher intakes of total protein, animal protein, and BCAAs and SCAAs were significantly associated with higher BMIz scores and fat mass. Additionally, there were significant positive associations between intakes of animal protein and SCAAs with WHtR. Not shown in the table, cysteine and methionine were also independently and positively associated with the outcome variables as when grouped into SCAAs. Fat-free mass showed significant associations with intakes of total and animal protein and both groups of amino acids, but this significance disappeared after further adjustment of the model for confounding variables. There were no significant associations observed between intakes of protein and the amino acids and height z-scores (Htz). Additionally, although plant protein intake was not significantly associated with any of the outcome measures, it was the only exposure variable that tended to show an inverse association with WHtR and a positive association with Htz in the fully adjusted model (see Table 5).

## 4. Discussion

This cross-sectional study is one of the few that looked into the associations of intake of protein and specific amino acid groups, primarily, with general body and central obesity, and secondarily, with body composition (fat mass and fat-free mass) among adolescents. Our findings show that intakes of total and animal proteins and BCAAs (leucine, isoleucine, and valine) and SCAAs (methionine and cysteine) were positively associated with general body adiposity and fat mass among our adolescent population. Higher intakes of animal protein and SCAAs were also associated with abdominal obesity.

Our results on total and animal protein intake and adiposity are consistent with those of Hermanussen [33], who reported a significant association between BMI standard deviation scores (BMI-SDS) and the mean absolute intake of all protein (r = 0.143, *p* < 0.0001) and animal protein (r = 0.151, *p* < 0.0001) in German adolescents of both sexes. Energy intake from protein was positively associated with BMI and waist-to-height ratio in a population of 8- to 12-year-old children [55] and with BMI z-score and % body fat among a cross-sectional sample of adolescents in the HELENA Study [34]. In a prospective study among adults who participated in the Diet, Genes, and Obesity (Diogenes) project of the European Prospective Investigation into Cancer and Nutrition, intakes of total protein and protein from foods of animal origin were found to be positively associated with weight gain, but not waist circumference changes. Plant protein, however, was not associated with either general body obesity or central adiposity [31]. Another prospective study among adults found animal protein intake to be associated with increased risk, and plant protein with lower risk, of obesity [28]. Similar results were found in a cross-sectional study among Belgians, which found plant protein to be inversely associated with overweight/obesity, while animal protein was only associated with increased obesity risk among men [56].

Among the overweight/obese subjects in our study, overall consumption of total protein, animal protein, and BCAAs and SCAAs were higher, while plant protein intakes were lower compared to their nonobese counterparts. The higher animal protein intake among the overweight/obese could be attributed to meat and poultry intake, which was found to be higher in this group. In a systematic review of studies on adults in both developed and developing countries, red and processed meat intake was found to be directly associated with a higher risk of obesity, BMI, and waist circumference [32]. In the GINIplus and LISAplus study that followed up children through adolescence, red meat exposure during childhood was found to be associated with fat mass during adolescence [57]. Our cross-sectional analysis indicates that the association between animal protein intake and fat mass could be due to intake of meats, and thus, could lend credence to the findings of the cited prospective study. It is noteworthy, however, to note that although no significant associations were observed between intake of plant protein and any of the outcome variables in the study, there was a tendency for an inverse relationship between plant protein intake and central adiposity, which remained the same after controlling for additional potential confounders. This tendency could be explained by the significantly higher intakes of plant protein and legumes (soy and non-soy) among those with normal weight and waist circumference.

Our adolescents with central obesity also had a slightly lower intake of dairy, while obese individuals had a higher consumption compared to those in the normal category. Lower intake of dairy in abdominally obese adolescents was previously reported [58] in a representative sample of U.S. adolescents. However, given the differences in our approaches to measuring central adiposity (waist circumference vs. waist-to-height ratio), comparing our results to those in that study might not be entirely appropriate.

Past investigations examined possible mechanisms on how animal protein intake can potentially increase the risk of obesity. Although unlikely to be the single contributor to obesity, red meat and its products are energy-dense foods and this could explain their association with overweight/obesity [14,32]. Meats are frequently consumed in the western diet, particularly by adolescents, and therefore, the magnitude of the effect on their health can potentially be significant. The positive link between animal protein intake could be related to the possible enhancement in stimulation of insulin [59] and insulin-like growth factor-1 (IGF-1) [60]. Although IGF-1 has major roles in the regulation of human growth, it has also been linked to adipocyte proliferation and differentiation [61]. On the other hand, a plant-based diet (in particular vegan diet) has been found to be associated with lower circulating levels of total IGF-1 and higher levels of IGF-binding proteins, suggesting a lower intake of protein high in essential amino acids and lower levels of IGF-1 [62]. An emerging line of evidence suggests that persistent organic pollutant content of foods of animal origin is contributing to an increased incidence of some lifestyle-related diseases including obesity [63]. Finally, emerging research indicates a potential link between higher levels of certain gut microbacteria and a higher occurrence of obesity [64]. This line of research, however, is still in early stages and therefore cannot be used as concrete evidence for verification of theoretical mechanisms.

Our significant positive associations between intake of BCAAs and higher weight and fat mass are not consistent with previous findings [39,40], which found strong inverse associations between the higher intake of dietary BCAAs and obesity and insulin resistance in samples of middle-aged adults. Other studies have shown associations of higher plasma BCAA concentrations with pediatric obesity [38] and insulin resistance [38,65] among adolescents. Recent reports show that BCAA is also independently associated with cardiovascular mortality [66] and cardiometabolic risk [67]. Furthermore, we found SCAAs to be positively associated with BMIz, WHtR, and fat mass, which remained significant even after controlling for several other possible confounders. Our results confirm the findings of a study done on Chinese adults, which showed an association between SCAA intake and BMI and waist circumference [68]. For both of these amino acid groups, we only estimated the dietary intake. Whether or not plasma concentration of BCAAs can be fairly represented by dietary intake is not known. Being one of the very few studies that investigated the relationship of BCAA and SCAA intake with obesity in the adolescent population, further investigations in this population would be needed to elucidate the role of these amino acids in adolescent health.

### Strengths and Limitations of the Study

One strength of our study is the absence of lifestyle habits that have a direct impact on health such as smoking and alcohol use in our study population. Our population also had a wide variation in intake of both plant and animal proteins and their food sources, thus increasing our power to find significant associations.

However, we also recognize a number of limitations: Our study sample did not represent the typical U.S. adolescent population. Overweight/obesity prevalence in this group was lower and the majority had highly educated parents. Thus, the results have limited generalizability. Dietary intake was self-reported in a food frequency questionnaire, which is subject to misreporting bias, particularly in the adolescent population [69]. However, validated food frequency questionnaires continue to be the mainstay in diet assessment for epidemiological studies. Another limitation was the inability to establish causal relationships due to the lack of temporality in a cross-sectional study. This study, however, is one of the very few that investigated protein–obesity relationships in an adolescent population and thus, can provide additional hypothesis-generating data. We did not take into account the proteins from fruits and vegetables that could have potentially contributed substantially to the total, plant protein, and amino acids intake estimates due to the high intake of such foods in this population. However, we considered it safe to assume that fruits and vegetables would have minimal contribution in protein intake and thus decided a priori not to include them. Because of a lack of data on dietary supplements [70], particularly protein supplements and BCAAs, our estimates of protein intake may be lower. However, an item in our questionnaire about dieting or use of special diet did not indicate the use of body-building substances/nutrients to enhance body image.

## 5. Conclusions

Our findings show that higher dietary intakes of total protein, animal protein, BCAAs, and SCAAs were significantly associated with general body adiposity and fat mass in this healthy adolescent population. We also found positive associations between intakes of animal protein and SCAAs with abdominal obesity. Adolescents with higher BMIz had a higher intake of total protein and animal protein from all the animal protein foods and lower intake of all the plant food with the exception of grains. Intake of the individual BCAAs and SCAAs was slightly higher in obese individuals. Our findings suggest that high dietary total and animal protein intake and BCAAs and SCAAs may be detrimental to the health of this adolescent population. Future longitudinal studies should be conducted to investigate these exposure variables as potential contributing factors to the overweight/obesity epidemic. In addition, given the results of our study, the long-term safety of high-protein diets should be evaluated in relation to the adolescent population.

## Figures and Tables

**Table 1 nutrients-12-00110-t001:** Demographic characteristics and anthropometric/body composition measurements, macronutrient intake and lifestyle habits of participants, collectively and by gender groups.

	All	Girls	Boys	*p* *
	*n* (%)	*n* (%)	*n* (%)	
All participants	530 (100)	299 (56.4)	231 (43.6)	
Age group (years)				0.772
12–13	126 (24.3)	69 (23.8)	57 (24.9)	
14–18	393 (75.7)	221 (76.2)	172 (74.1)	
Ethnicity ^a^				0.598
Caucasian	190 (37.8)	101 (35.9)	89 (40.3)	
Hispanic	72 (14.3)	39 (13.9)	33 (14.9)	
African/African American	47 (9.4)	24 (8.5)	23 (10.4)	
Asian	59 (11.7)	33 (11.7)	26 (11.8)	
Other	36 (7.2)	22 (7.8)	14 (6.3)	
Mixed	98 (19.5)	62 (22.1)	36 (16.3)	
Mother’s educational level				0.971
High School or less	74 (14.7)	42 (14.9)	32 (13.5)	
Some College or College Graduate	240 (47.8)	135 (48.0)	105 (47.5)	
Graduate level	188 (37.5)	104 (37.0)	84 (38.0)	
Father’s educational level				0.114
High School or less	91 (18.1)	44 (15.7)	47 (21.3)	
Some College or College Graduate	188 (37.5)	115 (40.9)	73 (33.0)	
Graduate level	223 (44.4)	122 (43.4)	101 (45.7)	
Site				0.249
California	289 (55.7)	155 (53.4)	134 (58.5)	
Michigan	230 (44.3)	135 (46.6)	95 (41.5)	
Dietary status ^b^				0.013
Vegetarian	137 (26.4)	89 (30.7)	48 (21.0)	
Nonvegetarian	382 (73.6)	201 (69.3)	181 (79.0)	
BMIz ^c^				0.001
Normal weight	405 (78.0)	224 (77.2)	181 (79.1)	
Overweight	92 (17.7)	59 (20.3)	33 (14.4)	
Obese	22 (4.2)	7 (2.4)	15 (6.6)	
Waist-to-Height Ratio ^d^				0.004
Normal	412 (79.4)	217 (74.8)	195 (85.2)	
Obese	107 (20.6)	73 (25.2)	34 (14.8)	
	**Mean (SD)**	**Mean (SD)**	**Mean (SD)**	***p*** ******
Age	15.0 (1.7)	15.0 (1.8)	15.0 (1.7)	0.889
Weight, kg	60.0 (14.1)	57.6 (12.6)	62.9 (15.2)	<0.0001
Height, cm	165.3 (9.3)	161.5 (7.0)	170.2 (9.5)	<0.0001
Weight-for-age z	0.33 (0.97)	0.35 (0.91)	0.31 (1.0)	0.647
Height-for-age z	0.11 (0.99)	0.06 (1.0)	0.16 (0.97)	0.251
BMIz	0.26 ± 0.99	0.34 (0.88)	0.17 (1.1)	0.052
Waist-to-height ratio	0.46 ± 0.06	0.46 (0.06)	0.45(0.06)	0.002
Fat mass, kg	13.1 ± 8.5	15.6 (7.6)	9.9 (8.5)	<0.0001
Fat-free mass, kg	47.0 ± 9.4	41.8 (5.1)	53.4 (9.6)	<0.0001
Total energy intake, kcal/day	2145 (748)	2013 (677)	2311 (799)	<0.0001
Total fat intake, g/day	83.3 (32.0)	78.8 (29.8)	88.9 (33.8)	<0.0001
Total carbohydrate intake, g/day	275.2 (103.1)	259.2 (96.9)	295.5 (107.3)	<0.0001
Total protein intake, g/day	86.1 (34.1)	79.4 (29.5)	94.6 (37.5)	<0.0001
Physical activity (min/day)	31.8 (25.2)	28.1 (23.9)	36.6 (26.0)	<0.0001
Sleep (h/night)	7.7 (1.2)	7.5 (1.2)	8.0 (1.2)	<0.0001

Notes: ***** Comparison by Chi-square; ****** Comparison by independent *t*-test. **^a^** Other: Ethnicities not included among the specific categories; Mixed: Parents have different ethnicities. **^b^** Vegetarian: Defined as one who eats less than one (1) combined portion (i.e., <3 oz) of meat, meat derivatives, poultry, and fish per week. **^c^** Categories were based on body-mass-index-for-age (BMIz) z-scores for general obesity: Normal, −2.00 to 1.00; overweight, >1.00; obese, >2.00. **^d^** Categories were based on a waist-to-height ratio (WHtR) cutoff point of 0.5 to differentiate between normal and centrally obese.

**Table 2 nutrients-12-00110-t002:** Characteristics of participants according to categories of body weight and central adiposity.

	BMI z-Score ^a^				Waist-to-Height Ratio ^b^			
	Normal (*n* = 405)		Overweight/Obese (*n* = 114)		Normal (*n* = 412)		Obese (*n* = 107)	
	Mean	SD/95% CI	Mean	SD/95% CI	Mean	SD/95% CI	Mean	SD/95% CI
Age	15.0	1.8	14.8	1.6	15.0	1.8	15.0	1.7
Total energy intake, kcal/day	2161	726	2085	821	2173	741	2039	769
Total fat intake, g/day ^c^	**82.1 ***	**13.8**	**85.7 ***	**14.5**	82.3	14.1	84.9	13.7
Total carbohydrate intake, g/day ^c^	**276.7 ***	**37.9**	**263.9 ***	**41.5**	275.1	39.9	269.1	35.5
Physical activity, min/day	32.4	25.6	30.0	24.0	**34.0 ***	**26.0**	**24.0 ***	**21.0**
Sleep, h/night	7.8	1.2	7.7	1.2	7.8	1.2	7.7	1.3
Fat mass, kg ^d^	**9.7 ****	**9.2–10.3**	**23.4 ****	**22.4–24.5**	**10.2 ****	**9.6–10.8**	**23.7 ****	**22.5–25.0**
Fat-free mass, kg ^d^	**46.1 ****	**45.5–46.7**	**53.0 ****	**52.0–54.1**	**46.7 ****	**46.1–47.3**	**51.6 ****	**50.4–52.9**
	***n***	**%**	***n***	**%**	***n***	**%**	***n***	**%**
All participants								
Gender								
Male	181	44.7	48	42.1	195	47.3	34	31.8
Female	224	55.3	66	57.9	217	52.7	73	68.2
Ethnicity (child) ^e^								
African/African American	32	8.2	15	13.5	35	8.8	12	11.5
Caucasian	154	39.4	36	32.4	161	40.5	29	27.9
Hispanic	56	14.3	16	14.4	56	14.1	16	15.4
Asian	50	12.8	9	8.1	49	12.3	10.0	9.6
Other	24	6.1	12	10.8	26	6.5	10.0	9.6
Mixed	75	19.2	23	20.7	71	17.8	27	26.0
Dietary Status ^f^								
Vegetarian	115	28.4	22	19.3	114	27.7	23	21.5
Nonvegetarian	290	71.6	92	80.7	298	72.3	83	78.5
Site								
California	233	57.5	56	49.1	243	59.0	46	43.0
Michigan	172	42.5	58	50.9	169	41.0	61	57.0

Notes: Values in bold indicate that normal and overweight/obese groups are significantly different: **p* < 0.05; ******
*p* < 0.001. **^a^** Categories of BMIz scores were determined using cutoff points for normal weight (BMIz < 1.00), overweight (BMIz > 1.00), and obese (BMIz > 2.00). **^b^** Categories of WHtR were determined using the cutoff point of 0.5 to differentiate between normal and abdominally obese. ^c^ Energy-adjusted values. **^d^** Estimated marginal means (95% confidence interval), adjusted for age and gender. **^e^** Other: Ethnicities not included among the specific categories; Mixed: Child with parents of different ethnicities. **^f^** Vegetarian is defined as intake of less than one combined portion of meat, meat derivatives, poultry, and fish per week.

**Table 3 nutrients-12-00110-t003:** Estimated energy-adjusted ^a^ means (SD) of total, animal and plant proteins, branched-chain amino acids (BCAAs), and sulfur-containing amino acids (SCAAs) intake and percent contribution to total protein intake according to gender, weight status, and central adiposity.

	Total Protein (g/day)	Total Protein (g/kgBW/day)	Animal Protein (g/day)	Plant Protein (g/d)	BCAA (g/day)	SCAA (g/day)	Protein as % Energy	Animal Protein	Plant Protein	BCAA	SCAA
	Energy-Adjusted Mean (SD)							Intake as % (SD) of Total Protein			
All	85.6 (14.7)	1.5 (0.40)	39.7 (20.6)	45.9 (16.1)	14.3 (2.8)	2.9 (0.7)	16.0 (2.5)	45.1 (18.2)	54.9 (18.2)	16.6 (0.90)	3.4 (0.37)
Gender											
Females	**84.4 (13.4) ***	1.5 (0.37)	37.6 (19.5)	46.8 (15.6)	**14.1 (2.6) ***	**2.0 (0.6) ***	**15.8 (2.4) ***	46.6 (17.6)	53.5 (17.6)	16.6 (0.8)	3.3 (0.3)
Males	**87.3 (16.0) ***	1.5 (0.43)	42.4 (21.8)	44.8 (16.7)	**14.6 (3.2) ***	**2.1 (0.8) ***	**16.3 (2.6) ***	43.9 (18.7)	56.1 (18.7)	16.7 (0.9)	3.4 (0.3)
BMIz ^b^											
Normal	85.0 (14.0)	1.6 (0.37)	**37.9 (20.3) ****	**47.2 (16.0) ***	**14.2 (2.8) ***	**2.0 (0.7) ***	**15.9 (2.4) ***	**43.4 (19.9) ****	**56.6 (19.9) ****	16.6 (0.8)	**3.4 (0.3) ***
Overweight/Obese	**88.0 (16.9)**	**1.2 (0.35)**	**45.9 (21.5) ****	**42.0 (16.1) ***	**14.8 (3.1) ***	**2.2 (0.7) ***	**16.5 (2.9) ***	**51.2 (18.5) ****	**48.8 (18.5) ****	16.8 (0.8)	**3.5 (0.3) ***
WHtR ^c^											
Normal	85.7 (15.0)	1.6 (0.40)	38.9 (21.3)	**46.8 (16.5) ***	14.3 (2.9)	2.0 (0.7)	16.0 (2.5)	**44.1 (18.5) ***	**56.0 (18.5) ***	16.6 (0.8)	**3.4 (0.3) ***
Obese	85.5 (13.8)	1.2 (0.30)	42.3 (18.4)	**43.1 (14.9) ***	14.3 (2.6)	2.1 (0.6)	16.0 (2.6)	**48.5 (17.4) ***	**51.5 (17.4) ***	16.7 (0.8)	**3.5 (0.3) ***

Notes: Values in bold indicate that normal and overweight/obese groups are significantly different: *****
*p* < 0.05; ******
*p* < 0.001. Differences were determined using independent *t*-tests. g/d, gram per day; g/kgBW/d; gram per kilogram body weight per day; BCAAs, branched-chain amino acids; SCAAs, sulfur containing amino acids; BMIz, body-mass-index-for-age z-score; WHtR, waist-to-height ratio. **^a^** Mean intake of all proteins and amino acid groups were energy-adjusted using the residual method. **^b^** Categories of BMIz scores were determined using cutoff points for normal weight (BMIz < −1.00), overweight (BMIz > 1.00), and obese (BMIz > 2.00). **^c^** Categories of WHtR were determined using the cutoff point of 0.5 to differentiate between normal and abdominally obese.

**Table 4 nutrients-12-00110-t004:** Estimated energy-adjusted means (SD) of animal and plant protein foods intake in g/d according to gender, weight status, and central adiposity.

Characteristic	Energy-Adjusted ^a^ Mean (SD) of Protein Foods Consumed in Grams/Day										
	Red Meat	Poultry	Processed Meat	Fish	Dairy	Egg	Grains	Gluten	Soy	Non-Soy Legumes	Nuts
All	6.9 (6.6)	5.8 (8.7)	1.0 (1.6)	1.2 (2.2)	17.5 (8.4)	2.6 (2.1)	16.7 (4.7)	2.0 (2.7)	10.4 (10.0)	3.6 (3.0)	3.4 (3.2)
Gender											
Females	**6.3 (6.3) ***	5.9 (8.2)	0.9 (1.6)	1.1 (2.1)	17.0 (8.1)	2.5 (2.1)	**16.2 (4.5) ***	2.9 (0.17)	10.8 (9.9)	3.6 (2.7)	3.4 (3.4)
Male	**7.8 (7.0) ***	5.8 (7.2)	1.2 (1.5)	1.3 (2.4)	18.1 (8.7)	2.7 (2.1)	**17.2 (4.9) ***	2.4 (0.15)	9.9 (10.1)	3.5 (3.3)	3.4 (3.1)
BMIz ^b^											
Normal	**6.4 (6.4) ***	**5.2 (6.5) ***	1.0 (1.5)	1.2 (2.4)	17.3 (8.3)	2.5 (2.1)	16.8 (4.6)	2.1 (2.4)	**11.1 (10.1) ***	**3.8 (3.2) ****	3.5 (3.3)
Overweight/Obese	**8.3 (6.7) ***	**8.2 (11.1) ***	1.2 (1.8)	1.4 (1.8)	18.4 (8.9)	2.8 (2.3)	16.0 (4.8)	2.0 (3.6)	**8.3 (9.6) ***	**2.9 (2.3) ****	3.1 (3.3)
WHtR ^c^											
Normal	6.6 (6.5)	5.5 (7.5)	0.99 (1.6)	1.2 (2.4)	17.6 (8.2)	2.5 (2.1)	16.7 (4.8)	2.1 (2.5)	**11.0 (10.5) ***	**3.8 (3.2) ***	3.4 (3.1)
Obese	7.5 (6.6)	7.0 (8.9)	1.2 (1.7)	1.2 (1.7)	17.2 (9.1)	2.8 (2.0)	16.3 (4.1)	1.9 (3.5)	**8.4 (8.0) ***	**2.9 (2.3) ***	3.5 (3.8)

Notes: Values in bold indicate that normal and overweight/obese groups are significantly different: *****
*p* < 0.05; ******
*p* < 0.001. Differences were determined using independent *t*-tests. BMIz, body-mass-index-for-age z-score; WHtR, waist-to-height ratio. **^a^** Mean intake of all protein foods were adjusted for energy using residual method. Sum of intake of protein from animal and plant sources do not add up to total plant protein intake because protein values for fruits and vegetables are not accounted for. **^b^** Categories of BMIz scores were determined using cutoff points for normal weight (BMIz < 1.00), overweight (BMIz > 1.00), and obese (BMIz > 2.00). **^c^** Categories of WHtR were determined using the cutoff point of 0.5 to differentiate between normal and abdominally obese.

**Table 5 nutrients-12-00110-t005:** Associations **^a^** between dietary protein and amino acid intake and weight status, central adiposity, and body composition.

Nutrient		BMIz		Waist-to-Height Ratio ^b^		Fat Mass ^c^		Fat-Free Mass ^c^		Htz	
	Model	β	95% CI	β	95% CI	β	95% CI	β	95% CI	β	95% CI
Total Protein ^d^ (per 10g/day)	Base	**0.105 *****	(0.047, 0.164)	0.015	(−0.001, 0.031)	**0.048 ***	(0.011, 0.084)	**0.009 ***	(0.001, 0.017)	0.001	(−0.056, 0.059)
	Full	**0.101 ****	(0.041, 0.161)	0.013	(−0.003, 0.029)	**0.044 ***	(0.007, 0.081)	0.008	(−0.0004, 0.016)	−0.005	(−0.064, 0.054)
Animal Protein ^d^ (per 10g/day)	Base	**0.117 *****	(0.058, 0.175)	**0.017 ***	(0.002, 0.033)	**0.051 ****	(0.015, 0.087)	**0.010 ***	(0.002, 0.018)	0.002	(−0.056, 0.061)
	Full	**0.118 *****	(0.057, 0.178)	**0.017 ***	(0.001, 0.033)	**0.049 ***	(0.011, 0.087)	0.008	(−0.0001, 0.016)	−0.01	(−0.070, 0.050)
Plant Protein ^d^ (per 10g/day)	Base	0.018	(−0.056, 0.093)	−0.003	(−0.023, 0.017)	0.016	(−0.031, 0.062)	0.003	(−0.008, 0.013)	−0.004	(−0.079, 0.070)
	Full	0.027	(−0.049, 0.103)	−0.003	(−0.023, 0.018)	0.021	(−0.026, 0.069)	0.006	(−0.005, 0.016)	0.019	(−0.056, 0.094)
BCAAs ^d^ (per 1g/day)	Base	**0.058 *****	(0.028, 0.088)	0.008	(−0.000, 0.016)	**0.025 ****	(0.006, 0.043)	**0.005 ***	(0.001, 0.009)	0.003	(−0.027, 0.033)
	Full	**0.056 *****	(0.025, 0.087)	0.008	(−0.001, 0.015)	**0.023 ***	(0.003, 0.042)	0.004	(−0.000, 0.008)	−0.002	(−0.030, 0.028)
SCAAs ^d^ (per 1g/day)	Base	**0.026 *****	(0.001, 0.004)	**0.005 ****	(0.001, 0.008)	**0.011 ****	(0.004, 0.019)	**0.002 ***	(0.000, 0.004)	0.000	(−0.012, 0.013)
	Full	**0.025 *****	(0.012, 0.038)	**0.004 ***	(0.000, 0.008)	**0.010 ****	(0.002, 0.018)	0.002	(−0.000, 0.003)	−0.003	(−0.015, 0.010)

Notes: CI, confidence interval; BCAAs, branched-chain amino acids; BMIz, body-mass-index-for-age z-score; Htz, height-for-age z-score. Values in bold indicate significant associations; *****
*p* < 0.05; ******
*p* < 0.01; *******
*p* < 0.001. ^a^ Associations determined using multiple regression analysis; Base model adjusted for age, gender, site, ethnicity, and total energy; Full model adjusted for age, gender, site, ethnicity, total energy, total fat, physical activity, and hours of sleep. ^b^ Inverse transformation applied; ^c^ Log transformation applied; ^d^ Energy-adjusted by residual model.

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
