# Peer review of "Animal Protein Intake Is Associated with General Adiposity in Adolescents: The Teen Food and Development Study"

_nutrients, 2019, doi:10.3390/nu12010110_

Round 1
Reviewer 1 Report
In the manuscript entitled “Animal protein intake is associated with general adiposity in adolescents: The Teen Food and Development Study”, Segovia-Siapco et al. present results of a study focused on investigating the role of protein intake in adolescent health. Obesity was defined as BMIz scores and waist-to-height ratios (WHtR). Association between obesity and intakes of total protein, animal protein, BCAA, SCAAs was evaluated. Based on these, authors suggest that high intake of total protein, animal protein, BCAA, and SCAAs is positively associated with body adiposity and fat mass. However, there are several issues that should be supported by additional experiments and/or discussion.
As reviewed in the part of introduction and discussion, the association between protein especially animal protein and obesity and the correlation between BCAA/SCAAS and obesity has been reported. What is novel finding? Please add information about normal and obese based on fat mass in Table 2, Table 3, and Table 4. Authors used 151-item semi-quantitative food frequency questionnaire. I wonder how to calculate/quantify BCAA and SCAAs from FFQ. To investigate a role of BCAA and SCAAs in central obesity/obesity, multivariate logistic regression analysis must be conducted to obesity odd ratios and corresponding 95% confidence intervals (Cis) for the risk of obesity.Author Response
Below are our responses to the points raised by REVIEWER 1. Thank you for taking the time to review our manuscript. We hope that our responses have adequately met the intended improvements.
In the manuscript entitled “Animal protein intake is associated with general adiposity in adolescents: The Teen Food and Development Study”, Segovia-Siapco et al. present results of a study focused on investigating the role of protein intake in adolescent health. Obesity was defined as BMIz scores and waist-to-height ratios (WHtR). Association between obesity and intakes of total protein, animal protein, BCAA, SCAAs was evaluated. Based on these, authors suggest that high intake of total protein, animal protein, BCAA, and SCAAs is positively associated with body adiposity and fat mass. However, there are several issues that should be supported by additional experiments and/or discussion.
As reviewed in the part of introduction and discussion, the association between protein especially animal protein and obesity and the correlation between BCAA/SCAAS and obesity has been reported.
What is novel finding?RESPONSE: Our study confirms findings from previous research, the study was done on an adolescent population with a substantial proportion that subscribe to plant-based diet (and thus, a wide range of protein types). We used an innovative validated 151-item web-based food frequency questionnaire to estimate nutrient intake in adolescents. Our study provided noteworthy new information regarding associations between protein and amino acids intake and obesity.
Please add information about normal and obese based on fat mass in Table 2, Table 3, and Table 4.RESPONSE: Table 2 already includes information about fat mass according to categories of the main outcome variables (central and general body adiposity). For this table, our objective was to show a few pertinent characteristics (select demographic, macronutrient intake, select lifestyle, and body composition—fat mass and fat-free mass) of participants who are categorized as either normal or overweight/obese in terms of known definitions based on BMIz and WHtR, and thus, would be appropriate. However, adding fat mass in the other two tables (Tables 3 and 4) will be inappropriate since there are no known validated cut-offs/thresholds to define obesity with body fatness (measured as either fat mass or % body fat). In fact, Ho-Pham et al. [1] criticized the inappropriate use of the 1995 WHO Technical Report as criteria for defining overweight/obesity in several existing studies. In these two tables (3 and 4) our purpose was to show the dietary intakes of protein types and food sources among those with and without obesity which we can only define with BMIz and WHtR. Thus, we cannot accede to the suggestion.
REFERENCE: Ho-Pham, L.T.; Campbell, L.V.; Nguyen, T.V. More on body fat cutoff points. Mayo Clin Proc 2011, 86, 584; author reply 584-585.
The authors used 151-item semi-quantitative food frequency questionnaire. I wonder how to calculate/quantify BCAA and SCAAs from FFQ.
RESPONSE: Thank you for this comment. In lines 112-115, we described how the protein and amino acid profiles of foods were determined with NDS-R. We included the term “amino acids in lines 114-115 and also added statements about how BCAAs and SCAAs were measured: Intake of BCAAs was determined by taking the sum of the total intakes of the branched-chain amino acids leucine, isoleucine and valine; SCAAs intake was measured as the sum of the total intake of the sulfur-containing amino acids methionine and cysteine. (now incorporated in lines 126-129)
To investigate a role of BCAA and SCAAs in central obesity/obesity, multivariate logistic regression analysis must be conducted to obesity odd ratios and corresponding 95% confidence intervals (Cis) for the risk of obesity.RESPONSE: Although logistic regression could have been used to determine associations between intake of BCAAs/SCAAs and central and general body obesity, we chose instead to do multiple regression analysis for three reasons: (1) our aim was to quantify the associations between the protein and amino acids intake with the 2 primary outcome variables—BMIz and WHtR—and 3 secondary variables—fat mass, fat-free mass, and Ht z scores. There are no known/established/defined categories for the secondary variables (please see our response to item #2), and thus, logistic regression analysis will not be appropriate for those variables; (2) logistic regression analysis is dependent on the number of cases; considering that we have a (healthy) study population with a smaller number of general body or centrally overweight/obese subjects compared to the general population, logistic regression will not be the best statistical analysis option; and, (3) the level of measurements of all our variables (both exposure and the outcome variables) are ratio level –we will lose information if we attempt to categorize our outcome variables. Please also remember that our exposure variables are not only BCAAs and SCAAs but total protein and the two types of proteins as well (animal and plant proteins).
Reviewer 2 Report
I generally agree with the conclusion made by the authors that animal protein intake is associated with adiposity, as presented by Table 5. However, I feel the presentation made here has been a bit biased by the preoccupation that "animal proteins are bad protein and plant proteins are good protein". I suggest the authors should described the results more carefully and discussed them more critically.
Major points
According to the BMI z-score in Table 2 and Table 3, the overweight/obese group had higher fat intake and lower carbohydrate intake than the normal group, while protein intakes did not differ significantly between the two groups. Table 4 indicates that the overweight/obese group had significantly higher animal protein intake than the control group. It should be looked at whether animal protein intake is associated with fat/carbohydrate intake, and it should be discussed whether the association, if any, is related to adiposity or not. Line 209-217; the authors explained a couple of differences in dietary intake between the normal and overweight/obese groups in Table 3, but most of them are statistically insignificant, meaning that they are not different. Only the difference we can see is that in protein as %Energy. Line 218-227; the authors described Table 4. While the data in Table 4 are expressed as g/d, the descriptions are made as %, which is a bit annoying and I hope this should be fixed. I cannot find the data of poultry in Table 4. Besides, in line 224-225, the authors mentioned "overweight/obese adolescents consumed higher amounts of all animal protein foods and lower amounts of all plant protein food", but most of these differences did not reach statistical significance. In the discussion section, the authors mainly focused on the similarity and discrepancies between the results of this study and those in previous reports. This is too plain to me, and I strongly suggest the authors should discuss why/how animal protein intake may lead to adiposity.Minor points
In Table 5, the figure "0.058" and "0.026" should be bold as they are statistically significant?Author Response
We would like to thank REVIEWER 2 for the time spent to review our manuscript. We found the feedback very helpful in improving the manuscript. Below are our responses to the points raised and hope that our responses have adequately met the intended improvements.
REVIEWER 2:
1. I generally agree with the conclusion made by the authors that animal protein intake is associated with adiposity, as presented by Table 5. However, I feel the presentation made here has been a bit biased by the preoccupation that "animal proteins are bad protein and plant proteins are good protein". I suggest the authors should described the results more carefully and discussed them more critically.
RESPONSE: Thank you for this observation/comment. We have done revisions in the paper to avoid the impression of bias when discussing the results and findings of the study.
Major Points
2. According to the BMI z-score in Table 2 and Table 3, the overweight/obese group had higher fat intake and lower carbohydrate intake than the normal group, while protein intakes did not differ significantly between the two groups.
Table 4 indicates that the overweight/obese group had significantly higher animal protein intake than the control group. It should be looked at whether animal protein intake is associated with fat/carbohydrate intake, and it should be discussed whether the association, if any, is related to adiposity or not.
RESPONSE: Yes, animal protein intake is moderately associated with fat intake (r=0.343) and strongly inversely correlated with carbohydrate (r=-0.618). Our analyses (full method) included fat intake as a covariate. We ran separately the analyses, adding each of these variables in the full model, and the relationships basically remained the same (very similar β values and 95% CIs) between animal protein and BMIz and WHtR as well as for the other variables in Table 5. Thus, we kept the values for what we had originally (fat as covariate), although we corrected some typos in the table.
3. Line 209-217; the authors explained a couple of differences in dietary intake between the normal and overweight/obese groups in Table 3, but most of them are statistically insignificant, meaning that they are not different. Only the difference we can see is that in protein as %Energy.
RESPONSE: We apologize for the confusion. The variables that were mentioned were indeed statistically significant. We reviewed the table values to make sure that we didn’t miss any markings to show sig differences. The changes are now shown (please see Table 3) and Section 3.2 had been revised accordingly (please see lines 219-229).
4. Line 218-227; the authors described Table 4. While the data in Table 4 are expressed as g/d, the descriptions are made as %, which is a bit annoying and I hope this should be fixed. I cannot find the data of poultry in Table 4.
RESPONSE: In our attempt not to mention what is already shown in the table, we discussed the ranking of protein food sources in this group. However, we thank the reviewer for the observation that this is and confusing. The section that discusses Table 4 had been revised (please see Section 3.3, lines 231-238) and the table now shows poultry instead of “white meat” to be consistent with the narrative.
5. Besides, in line 224-225, the authors mentioned "overweight/obese adolescents consumed higher amounts of all animal protein foods and lower amounts of all plant protein food", but most of these differences did not reach statistical significance.
RESPONSE: Thank you for pointing out this error. The necessary changes were made in Section 3.3 (please see lines 236-238).
6. In the discussion section, the authors mainly focused on the similarity and discrepancies between the results of this study and those in previous reports. This is too plain to me, and I strongly suggest the authors should discuss why/how animal protein intake may lead to adiposity.
RESPONSE: Thank you for this suggestion. In the discussion section, we have added a paragraph (lines 312-327) to provide possible explanations for the associations between animal protein intake and obesity. We have stated:
Past investigations examined possible mechanisms on how animal protein intake can potentially increase the risk of obesity. Although unlikely to be the single contributor to obesity, red meat and its products are energy-dense foods and this could explain their association with overweight/obesity [32, 59]. Meats are frequently consumed in the Western diet, particularly by adolescents, and therefore the magnitude of effect on their health can potentially be significant. The positive link between animal protein intake could be related to the possible enhancement in stimulation of insulin [60] and insulin-like growth factor-1 (IGF-1) [61]. Although IGF-1 has major roles in regulation of human growth, it has also been linked to adipocyte proliferation and differentiation [62]. On the other hand, a plant-based diet (in particular vegan diet) has been found to be associated with lower circulating levels of total IGF-1 and higher levels of IGF-binding proteins, suggesting a role of lower intake of protein high in essential amino acids and lower levels of IGF-1 [63]. An emerging line of evidence suggests that persistent organic pollutants content of foods of animal origin are contributing to an increased incidence of some lifestyle-related diseases including obesity [64]. Finally, emerging research is indicating a potential link between higher levels of certain gut microbacteria and higher occurrence of obesity [65]. This line of research, however, is still in early stages and therefore cannot be used as concrete evidence for verification of theoretical mechanisms.
Minor points
In Table 5, the figure "0.058" and "0.026" should be bold as they are statistically significant?
RESPONSE: Thank you for pointing this oversight. Those values are now in bold (see Table 5).
Round 2
Reviewer 1 Report
No more question/comment
Reviewer 2 Report
No new concern.